# Climate variability and Germanic settlement dynamics in the Middle Danube region during the Roman Period (1st–4th Century CE)

Marek Vlach[1⊙]*, Balázs Komoróczy[1⊙], Max Carl Arne Torbenson[2,3⊙], Jan Esper[2,3⊙], Rudolf Brázdil[3,4⊙], Ulf Büntgen[3,4,5⊙], Daniela Semerádová[3,6‡], Otmar Urban[3⊙], Jan Balek[3,6‡], Tomáš Kolář[3,7‡], Michal Rybníček[3,7‡], Miroslav Trnka[3,6⊙]

1 Institute of Archaeology, Czech Academy of Sciences, Brno, Czech Republic, 2 Department of Geography, Johannes Gutenberg University, Mainz, Germany, 3 Global Change Research Institute, Czech Academy of Sciences, Brno, Czech Republic, 4 Department of Geography, Faculty of Science, Masaryk University, Brno, Czech Republic, 5 Department of Geography, University of Cambridge, Cambridge, United Kingdom, 6 Department of Agrosystems and Bioclimatology, Faculty of Agronomy, Mendel University in Brno, Brno, Czech Republic, 7 Department of Wood Science and Wood Technology, Mendel University in Brno, Brno, Czech Republic

⊙ These authors contributed equally to this work.
‡ These authors also contributed equally to this work.
* vlach@arub.cz

## Abstract

Climatic variability inevitably impacted past societies and acted as a driver of change. The combined analyses of the archaeological record and written documentary sources, together with high-resolution climate reconstructions, remain rare. In this work, we compare evidence of change at the Germanic settlements (residential areas) of Iron Age Germanic societies in the Middle Danube region (the region of Moravia in the Czech Republic, Lower Austria and the Záhorie region in Slovakia) and reconstruct the effect of agroclimatic conditions during the first four centuries CE. Based on data from 773 residential areas with temporal identification, we demonstrate a coherent relationship between spatiotemporal changes in Germanic settlement structures and agroclimatic conditions. A nearly exponential increase in settlement structure during the 1st and half of the 2nd century CE coincided with improved agroclimatic conditions, whereas the subsequent settlement structure decline during the Late Roman Period temporally overlapped with agroclimatic deteriorations. Documented peak in cessations of residential areas in the late 2nd century CE appears unrelated to regional agroclimatic conditions and was instead caused by the Marcomannic Wars. We argue that separating periods of agroclimatic importance and insignificance is the first step towards identifying possible causal environmental drivers of settlement dynamics and societal change in the Middle Danube region.

**Data availability statement:** All relevant data used in the paper are available from OSF database at https://osf.io/jp5kx/. All data on agroclimatic reconstructions are available from OSF database at https://osf.io/pz2ja.

**Funding:** U.B., T.K. and M.R. contributions were funded by the Czech Science Foundation (https://gacr.cz/en/) through grant 23-08049S, Central European HYDROclimate from Oak stable isotopes over the past 8000 years – HYDRO8. M.Torb., J.E., R.B., O.U., J.B. and M.Trn. were supported by the Johannes Amos Comenius Programme and the Ministry of Education, Youth and Sports of the Czech Republic (https://msmt.gov.cz/) for the project "AdAgriF – Advanced methods of greenhouse gases emission reduction and sequestration in agriculture and forest landscape for climate change mitigation" (CZ.02.01.01/00/22_00 8/0004635). The paper originated within the framework of a solution of the Czech Science Foundation (https://gacr.cz/en/) grant project no. 20-11070S Protohistoric Communities of the "Marcomannic" Settlement Zone in the Middle Danube Region - Structure and Dynamics on the Basis of Digital Modelling (M.V and B.K.). It also originated as a part of the activities within the NASSA project (Network for Agent-based Modelling of Socio-ecological Systems in Archaeology) no. W001220N, funded by the Research Foundation - Flanders (FWO) (https://www.fwo.be/nl/). The funders play no role in the study design, data collection and analysis, decision to publish, or preparation of the manuscript.

**Competing interests:** The authors have declared that no competing interests exist.

## Introduction

The development of past societies has always been influenced by a wide range of factors, amongst which short- and long-term changes in climatic conditions are considered key drivers. Many influential archaeological and historical theoretical constructs and models, therefore, stress the role of climate variability in shaping the evolution of these societies (e.g., [1–5]). The level of subsistence resilience and susceptibility is ultimately a complex outcome of various technological (methods and tools for food production), demographic (population size and health conditions; e.g., [6]), and socioeconomic (e.g., redistribution system properties, forms of land owner-ship; [7]) factors.

Identifying the causal relationships between climate variability and societal devel-opment requires comprehensively addressing various internal (anthropogenic) and external (environmental) factors. This allows us to distinguish structural correlations from coincidental local events. Simultaneously, the extent of climatic dependence is mitigated by various adaptation strategies constrained by the tradition of practices and technological capabilities (e.g., use of fallow, crop rotation, manuring, and the balance between crop cultivation and animal husbandry), which could have partly compensated unfavourable conditions for basic subsistence procurement. Therefore, the emerging trajectories of past human development patterns, reconstructed from archaeological and historical sources and natural scientific data, are the complex results of many individual factors and phenomena and drivers of internal and external nature.

During the first four centuries of the Common Era (CE), two profoundly different worlds had emerged in the Middle Danube region and created a unique borderland of coexistence with the Roman Empire (provinces of Pannonia and Noricum) on the one side and the Germanic settlement region to the north of the Danube Limes on the other. The low-lying areas to the west of the Lesser and White Carpathians (i.e., the present-day regions of Moravia in the Czech Republic, Lower Austria, and Slovakian Záhorie) were settled by the ancient Germanic population, conventionally associated with the 'tribal' entity known as Marcomanni (Fig 1) [8–10]. Their societal organisation exhibited characteristics of the advanced, complex, or paramount chiefdom [10–13], also associated with the so-called *early state analogues* [14]. This concept encom-passes specific cases where the political entity's population or territory size already complies with the state level, however, the organisation structure exhibits pre-state forms of advanced chiefdom.

The agricultural production of these societies can be broadly characterised as primarily autarkic-oriented subsistence at the Iron Age level [10,15,16], as is also corroborated by contemporary narrative sources [17]. It was bound to a large amount of relatively small rural settlements [18,19], i.e., aggregation of various areas of activ-ities, including stable habitation. In available data, only limited features visible in the archaeological record suggest stable 'central' places of political, military, or economic power such as fortification systems, strategic or elevated locations or representative building structures [18, 20-22]. Observable changes in settlement structure appeared

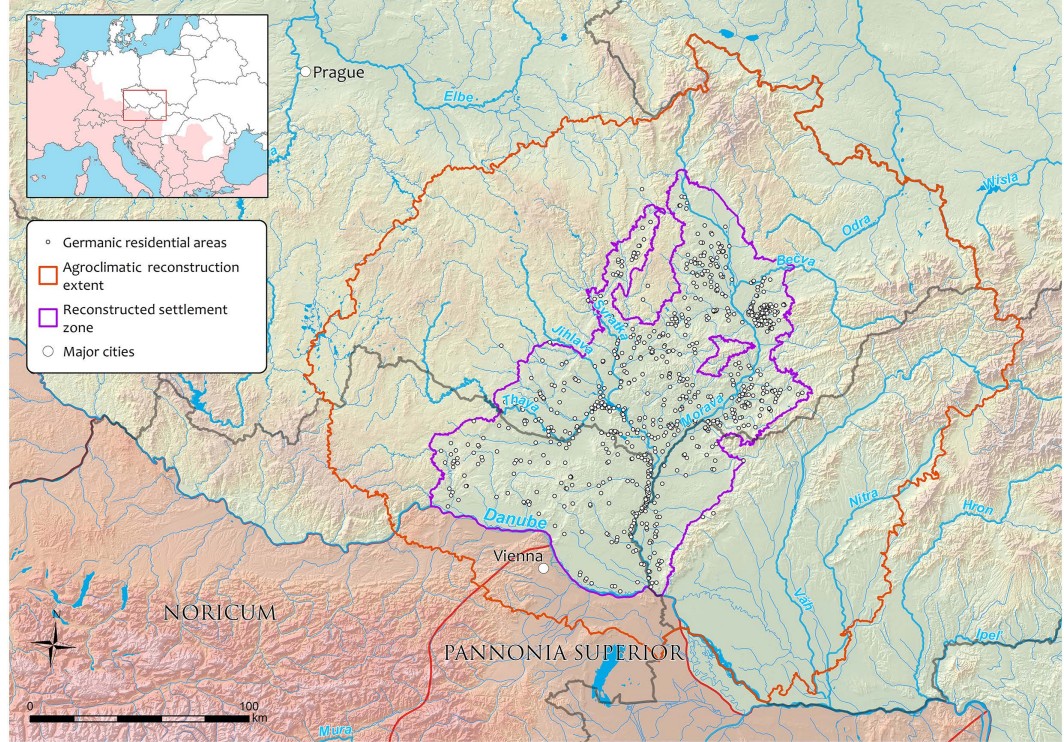

**Fig 1. Delimitation of the study region of the 'Marcomannic' settlement zone (violet line, [10]) with point representation of identified Germanic residential areas (white dots) bordering Roman provinces (pink area and red line) together with the area covered by agroclimatic reconstructions (brown line).**

only during the Late Roman Period (conventionally 200–400 CE in archaeology of the Germanic populations [23]), with the scant emergence of habitation at elevated locations [24]. Nevertheless, the growing basis of archaeological information, especially obtained through metal detector prospecting [25,26], allows for assuming certain forms of settlement hierarchy [27]. One such particularly apparent 'place of power' was found in the Mušov region (south Moravia) at the confluence of the regional rivers through outstanding archaeological evidence of residential and funerary nature [10,28]. However, the standard manifestation of Germanic habitation is generally uniform throughout the study period – a prerequisite for evaluating the relationship between archaeological and palaeoclimatic data.

Only recently have two unique datasets – palaeoclimatological [29,30] and archaeological [10,31,32] have made available evaluation of the extent and magnitude of climatic resilience and susceptibility of the Middle Danube Germanic societies. Their spatiotemporal overlap presents new opportunities for archaeoclimatic research. The pivotal aim of this study is to identify potential relationships and causations within these datasets, with an underlying assumption that the primarily agrarian-oriented economy of Germanic communities would respond, at least to some degree, to changes in climatic patterns.

## Materials and methods

### Archaeological data and documentary sources

The archaeological data within Central Europe contain a significant volume of information on the development of various aspects of the Germanic societies of the Roman Period. Despite a number of biases of qualitative and quantitative nature embedded in archaeological data [33], they provide indispensable sources for a better understanding of past societies.

Their full potential was not unlocked until recently, as the project-based activities of the Institute of Archaeology of CAS, Brno (Research Centre for Protohistoric Archaeology; See Acknowledgements) collected and synthesised all available archaeological data from a range of (foremost published) sources [10,31,32]. The emerging dataset MARCOMANNIA presently contains 87,500 records of both attribute (n = 52,100) and spatial nature (n = 35,400). This dataset provides a crucial basis for establishing probabilistic temporal proxies on the development of demographic, economic, societal and geopolitical aspects of the 'Marcomannic' settlement zone [10,31,32]. While acknowledging the existing biases in the quality of input information [18,34], the 773 available residential areas with associated spatiotemporal information provide substantial quantitative and qualitative input to reconstruct the general demographic development trends [35–37] of the local population [22,32], as well as serve as a basis for the archaeoclimatic analysis.

The archaeological knowledge of Germanic residential areas has been accumulating gradually over time [34], with a wide range of sources in terms of both quality and quantity [32]. The present state of knowledge, as documented in the MARCOMANNIA dataset [10], is indicative of the anticipated variation in reference-type resources, which are distributed exponentially. A substantial proportion of the documented residential complexes have been identified through surface surveys conducted in recent or distant past [34]. A significant increase in archaeological knowledge in recent decades can be attributed to systematic citizen science initiatives, primarily focused on the search for metal artefacts [25]. These activities have resulted in the substantial enrichment of archaeological knowledge in various domains, including typochronological and technological characteristics of material culture [10,31]. This has led to the discovery of new settlements and the refinement of the extent and chronological position of previously known sites [27,38].

An additional source of information for interpreting the archaeological data originates from the surviving ancient narrative sources [39–41]. Such records provide the dynamics and temporal resolution missing in archaeological sources [33]. Despite their concern for the neighbouring Germanic territories, often having episodic character and suffering by own biases, they provide indispensable insight into the wide range of qualitative and quantitative aspects of past Germanic societies (Tacitus [17]) or particular periods and events, such as the Marcomannic Wars (e.g., Cassius Dio [42], Herodian [43], *SHA* [44]) [45].

However, the embedded specifics of ancient narrative sources require critical consideration [13,46]. They tend to cover the studied temporal extent inconsistently and are burdened by various biases. The most prominent areas represent the political or military agenda of narrative (justification of particular approaches and means) including tendential exaggeration or underrating certain aspects or events [47]. A specific bias stems from focusing on a particular readership, mostly educated aristocrats and prominent persons. Moreover, the significant information filter is embedded in the references to spatially remote contexts, such as the ancient Germanic societies [13,48]. Consequently, the authors of these narratives were seldom direct observers of the events or phenomena they describe, which has a considerable impact on the credibility of the narratives (cp. [22,49,50]). Nevertheless, qualitative rather than quantitative aspects of the available narratives have been considered in assessing climate-related changes in the Germanic context of the study region.

## Agroclimatic and palaeoclimatic proxies

Wood from living, historical, and archaeological oak trees (*Quercus* spp.) in the Czech Republic, in which each tree ring represents an absolute-dated and annually resolved layer of growth increment, has been used to develop a multi-millennia tree-ring chronology covering the past 2,000 years [29]. Representation of tree-ring stable isotope ($\delta^{18}$O and $\delta^{13}$C) has been shown to be closely related to variations in local and regional climate [51]. Reconstructions of soil moisture [29], temperature sums, and water balance [52] have been produced from these tree-ring stable isotope chronologies. Such reconstructions of past climatic variability have been quantitatively and qualitatively verified against documentary evidence for the past 900 years and display a high degree of accuracy in capturing individual extreme years and prolonged periods of climatic anomalies. The reconstructions of annual temperature sums (sum of temperature for days > 10°C) and June-August water balance have been further used to estimate past agroclimatic potential [30], as they

relate to modern relationships between crops and weather, being formalised into the nine ordinal scale categories of "production regions" [53]: Very warm and dry (1), Warm and dry (2), Highly productive – dry (3), Highly productive – humid (4), Sufficiently productive – warm (5), Sufficiently productive – cold (6), Insufficiently productive – warm (7), Insufficiently productive – dry (8), Insufficiently productive – cold (9). These categories were quantified for the 1961–2020 instrumental period, with the reconstructed temperature sum and water balance estimates extrapolated across regional cadastre units based on the spatial relationship of the same period [30].

To evaluate the potential driving factors of settlement structure changes, other relevant palaeoclimatic proxies were used [54], foremost the Greenland ice core records and evidence of the past volcanic activities [40,55]. Specifically, for more significant eruptions with assumed weather extremes and effects on human societies [56], we consider events for which the GISP-2 ice core record displays $SO_4$ values exceeding 45 ppb (Fig 7 ) [40].

## Methods of analysis

Because of the inconsistent temporal resolution of the archaeological and palaeoclimatological datasets, 50-year time blocks have been selected to find the relationship between Germanic population development and changing agroclimatic potential. These time blocks have been used based on the mean value of phase duration [31,32]. For all the input residential areas, temporal probability distributions were derived through the principles of the aoristic calculation [57–59] and the presumed inceptions (hereon referred to as *Foundation*) and cessations (hereon referred to as *Abandonment*) of their primary function were deduced as either first or last nonzero value of aoristic probabilistic distributions through time block framework. The *Continuity* measure was established through the observed values between the time blocks.

A probabilistic simulation [10,58] was employed to enhance temporal resolution and provide more detailed insight into the development tendency of the Germanic settlement structure. The primary input consisted of the dichotomically expressed evidenced *Foundation* and *Abandonment*. This approach allows us to compensate two significant uncertainties in the archaeological record of the residential areas – the issue of precision of archaeological dating and the potential of an unevidenced hiatus in habitation [10]. The archaeological dating uncertainty was formalised as a more broadly dispersed likelihood of identification of their transitional events with a bilateral probability overlap of 25 years (half of a time block as well as 'standard' premodern mean life expectancy) [60] outside the particular time block, where an arbitrarily preset 33% probability of hiatus occurrence provides an additional variability in the simulation process. In tandem with the time block framework, it provides a more continuous temporal probabilistic distribution of the residential areas' development, which allows for the capture of trends that are not observable in the basic aoristic sum representation.

An integer array (ordinal scale) of the annual temporal and detailed spatial resolutions of the *Production region*s (from 1 to 9; Fig 6) has been generated for their spatial point distribution (Fig 1). Consecutively, the annual distribution of the *Production region* values has been aggregated into the time block framework, providing their average patterns (further as AVG) and temporal stability expressed by standard deviations (further as STD). The resulting dataset of aggregated agroclimatic reconstructions and documented residential areas is provided in OSF repository (see Data Availability Statement).

The main evaluation principle of the magnitude of the climate-driven development of the Germanic settlement structures is based foremost on an exploration of the incidence in the frequency of recorded *Foundation* and *Abandonment* and the improvement or deterioration of agroclimatic conditions based on the changes (difference) in *Production region* values in AVG and STD between two neighbouring time blocks. Corresponding differences were calculated as a simple subtraction of the value of a given time block from the value of a preceding block. A threshold value for the positive incidence in human response to the change of agroclimatic conditions was established as the average of the differences between neighbouring time blocks (±0.173 for AVG and ±0.015 for STD). In the anthropo-climatic impact evaluation, the time blocks represented individual temporal segments, where the incidence (climatic conditions improvement/deterioration and *Foundation*/*Abandonment* of the residential areas) was expressed through the percentage in the respective time block (Table 1).

**Table 1. The input archaeological data (upper part) with general probability distribution (Aoristic sum), totals of recorded transitional phases (*Foundation* and *Abandonment*, see Fig 2; also in percentages), and the data from agroclimatic reconstructions (central part) recorded in residential areas (*Production region* AVG and STD), followed by the percentages of recorded incidences in transitions and agroclimatic conditions between the time blocks (lower part). The values in rows represent individual percentages in each time block.**

| Data in Germanic residential areas | | −30–0 | 0-50 | 50-100 | 100-150 | 150-200 | 200-250 | 250-300 | 300-350 | 350-400 | 400-430 |
|---|---|---|---|---|---|---|---|---|---|---|---|
| **Archaeological data** | Aoristic sum | 4.5 | 32.2 | 64.1 | 149.5 | 148.1 | 110 | 110.1 | 71.5 | 58.5 | 21.0 |
| | *Foundation* (count) | 71 | 131 | 41 | 241 | 61 | 138 | 10 | 72 | 5 | 0 |
| | *Abandonment* (count) | 1 | 4 | 13 | 4 | 253 | 12 | 71 | 199 | 64 | 149 |
| | *Continuity* (count) | 0 | 73 | 200 | 228 | 465 | 271 | 397 | 336 | 208 | 149 |
| | *Foundation* (%) | 9.2 | 17 | 5.3 | 31.3 | 7.9 | 17.9 | 1.3 | 9.4 | 0.6 | 0 |
| | *Abandonment* (%) | 0.1 | 0.5 | 1.7 | 0.5 | 32.9 | 1.6 | 9.2 | 25.8 | 8.3 | 19.4 |
| **Agroclimatic data (*Production region*)** | AVG | 3.65 | 3.49 | 4.3 | 3.87 | 3.9 | 3.67 | 3.62 | 4.37 | 4.63 | 5.29 |
| | STD | 0.61 | 0.63 | 0.72 | 0.65 | 0.6 | 0.49 | 0.54 | 0.67 | 0.77 | 0.83 |
| **Incidence with *Foundation* (%)** | Improvement in AVG | 1.4 | 48.9 | 0 | 90.5 | 6.6 | 84.8 | 0 | 0 | 0 | 0 |
| | Improvement in STD | 70.4 | 31.3 | 12.2 | 66.8 | 60.7 | 90.6 | 20 | 11.1 | 0 | 0 |
| | Improvement in both | 1.4 | 16 | 0 | 60.6 | 1.6 | 75.4 | 0 | 0 | 0 | 0 |
| **Incidence with *Abandonment* (%)** | Deterioration in AVG | 0 | 0 | 100 | 0 | 12.6 | 0 | 0 | 99 | 85.9 | 99.3 |
| | Deterioration in STD | 100 | 25 | 61.5 | 0 | 24.9 | 8.3 | 56.3 | 75.9 | 95.3 | 58.4 |
| | Deterioration in both | 0 | 0 | 61.5 | 0 | 2.7 | 0 | 0 | 75.9 | 84.4 | 57.7 |

## Results

### Germanic settlement structure and agroclimatic susceptibility

The Germanic residential area data, expressed in 50/30-year time blocks (Fig 2), show an exponential increase during the 1st and the first half of the 2nd centuries CE, during which its spatial distribution throughout the 'Marcomannic' settlement zone reached its maximum. After flattening the trend during the second half of the 2nd century CE, the first significant drop was towards the first half of the 3rd century CE. After regaining stability in the second half of this century, the three consecutive decreases in the size of the settlement structure reached a minimum at the beginning of the 5th century CE [10]. Comparable quantitative development patterns could also be observed in the Germanic environments to the north of the study region associated with Wielbark and Przeworsk cultures [61], where the documented residential areas peak during the 2nd century CE. Generally similar trends also originate from other regions, including the wider context of the Roman Empire (e.g., [62–64]). Simultaneously, individual regions have their specific array of conditions, and distinctive effects of regionality have to be considered [65], with differentiated susceptibility towards the change of climatic conditions [66,67]. On the empire-wide level, many projections reflect the increase between the reconstructions for 14 and 165 CE, which underline generally significant overall increase between the 1st and 2nd centuries CE [65,68]. The sources of available information in individual regions vary and are biased either by qualitative or quantitative aspects. The resulting demographic reconstructions based on various types of proxies have differentiated temporal resolution (e.g., [62,69,70]). For instance, the individually explicit and annually resolved tax revenue records from Roman Egypt provide an exceptional source of information for various reconstructions on demographic aspects [71,72], where often multiple drivers participated in the image drawn from the administrative records [73]. Nevertheless, the comparable patterns could be, in general terms, also identified in more detailed and multiproxy-oriented studies, e.g., from central Italy [74], the Lower Rhine frontiers [63,75], or England [76]. Eventually, despite various approaches and reconstructions based on quantitatively substantiated archaeological and other relevant data being more frequent presently, they are still relatively scarce for broader comparisons [77].

The spatial distribution of residential areas of the study region (Fig 1) shows a preference for low-lying areas (generally below 350 m ASL), where proximity to water sources and high-quality soils played crucial roles in determining location

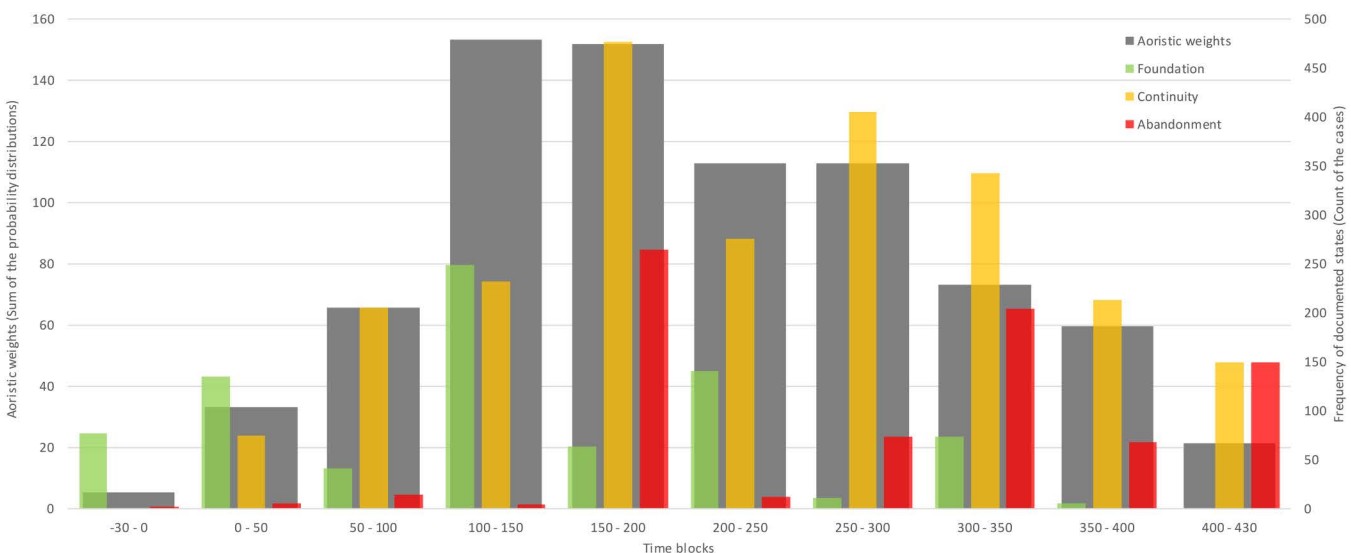

**Fig 2. Temporal distribution of the Germanic residential areas based on 50-year time blocks (grey) and terminal 30-year time blocks differenti-**
ated by *Foundation* (green), *Continuity* (orange), and *Abandonment* (red).

suitability [22]. The importance of local conditions is also apparent in the agroclimatic data, expressed by the *Production region* values (AVG and STD) in and near the areas of highest habitation density (Figs 4 and 5). The baseline average of 4.01 (i.e., a borderline between the *Production regions* 4 and 5), corresponding to the *Highly productive–humid* and *Highly productive–dry* categories, distinctively suitable for successful agricultural production, reflects their dominant representation in the land. Compared to the whole spatial agroclimatic domain, the residential areas lie outside less favourable either *Very hot and dry* or generally *Insufficiently productive* regions (warm, dry and cool) (Fig 3B).

The relationship between Germanic settlement activities and agroclimatic conditions was also apparent from the fact that 77% of the residential areas were 'active' during periods with more favourable agroclimatic conditions (lower AVG values of the reconstructed *Production regions*) than in their 'passive' phases. Similarly, 69% of the residential areas were 'active' during more stable agroclimatic conditions corresponding to the lower STD values in the reconstructed *Production regions*.

## Relationships between the settlement structure and agroclimatic potential

The residential area data exhibits three distinctive peaks (those with above 10% of all recorded positive or negative transitions for the whole study period; Table 1): both in the *Foundation* (0–50, 100–150, and 200–250 CE, i.e., 17%, 31%, and 18% of all recorded increases between time blocks) and *Abandonment* (150–200, 300–350, and 400–430 CE; i.e., 33%, 26% and 19% of all recorded decreases between time blocks) that suggest major changes in the settlement structure [10], either caused by internal or external drivers (or their combination), and which could be referred to as *key time blocks*. Therefore, these periods bear the highest explanatory potential for archaeoclimatic connection. Conversely, the time blocks with insignificant evidence of transitional events will likely provide limited insight into these relationships.

In the case of recorded *Foundations*, the largest peak was found for the time block 100–150 CE, where 90% of emerging residential areas experienced improved agroclimatic potential (AVG) and 67% experienced increased stability (STD) in agroclimatic conditions (Table 1). Two smaller peaks in Foundation occurred during 0–50 CE and 200–250 CE, with the latter showing a more significant improvement in agroclimatic conditions (85% AVG, 91% STD) compared to the earlier period (49% AVG, 31% STD). Overall, improved agroclimatic conditions (AVG, STD, or both) coincided with recorded

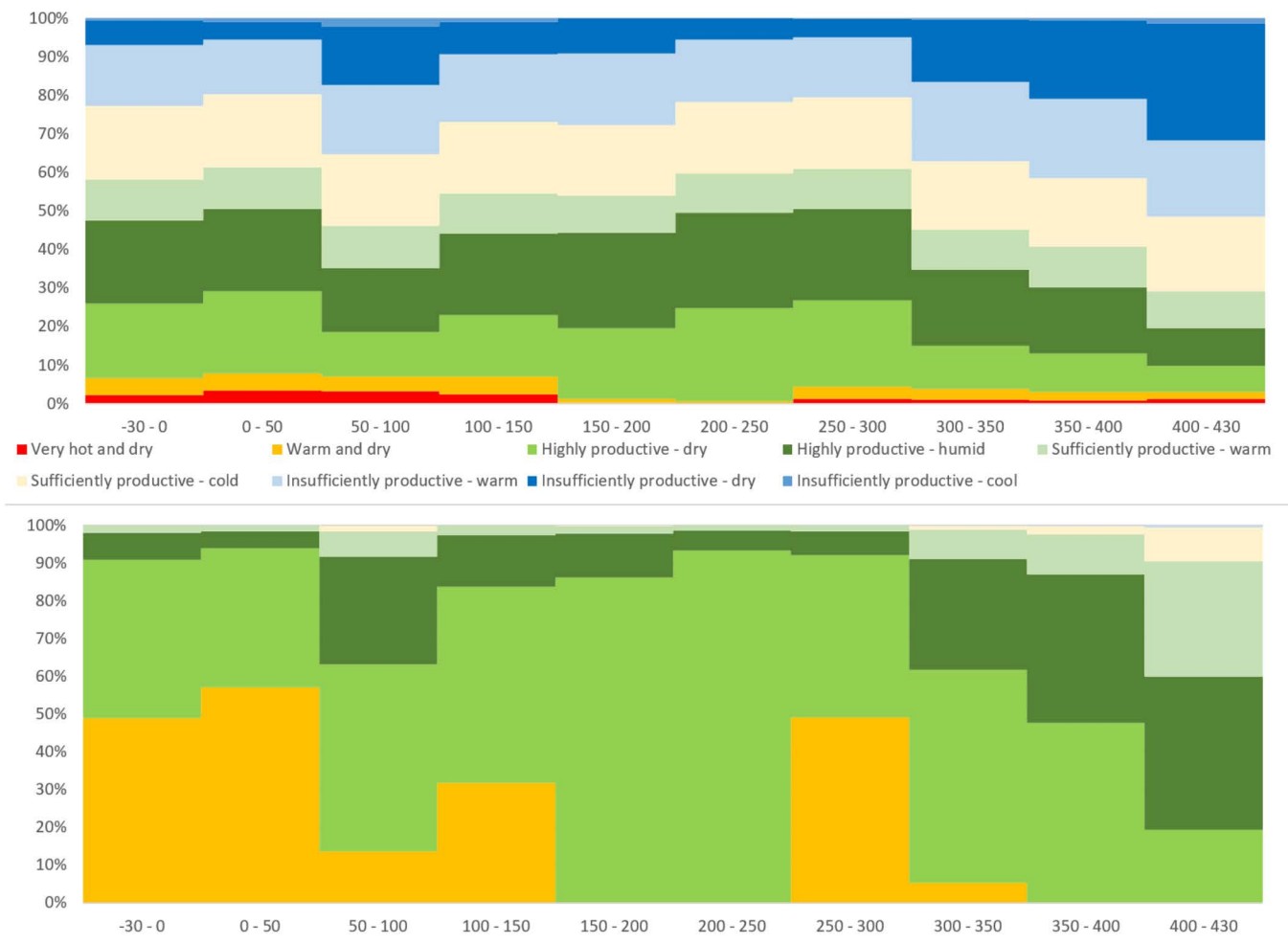

**Fig 3. Long-term changes of agroclimatic potential: A) proportions (%) of basic production areas across the full agroclimatic domain (brown line in Fig 1), and in B) the Marcomannic settlement zone (violet line in Fig 1) during the Roman Period.**

*Foundations* in 75% of all residential areas. A similar trend was also apparent for the *Abandonment* transitions. However, the highest peak in *Abandonment* during the *key time block* 150–200 CE cannot be related to a change in agroclimatic conditions (Fig 6a), likely explaining this upheaval in settlement structure by other factors, as suggested by both archaeological and historical sources (see Discussion).

Nevertheless, the consecutively highest *Abandonment* rates during the time blocks 300–350 and 400–430 CE display a considerable relationship between *Abandonment* and the deterioration of agroclimatic conditions, with 99% of residential areas showing declining AVG values during these periods. Additionally, 76% and 58% of them experienced increased variability in agroclimatic conditions, as indicated by higher STD values. For the whole study period, 76% of all *Abandonments* were associated with declines in agroclimatic conditions in either AVG, STD or both. Hence, in the *key time blocks*, a significant increase (decrease) in settlement activities generally coincided with the positive (negative) changes in agroclimatic conditions. The evidenced *Continuity* of the Germanic settlements inevitably coincided with the aoristic sum-based distribution (cp. Fig 2) with peaks in the time blocks 150–200 and 250–300 CE, which also pointed out the general stability in the settlement structure development and which

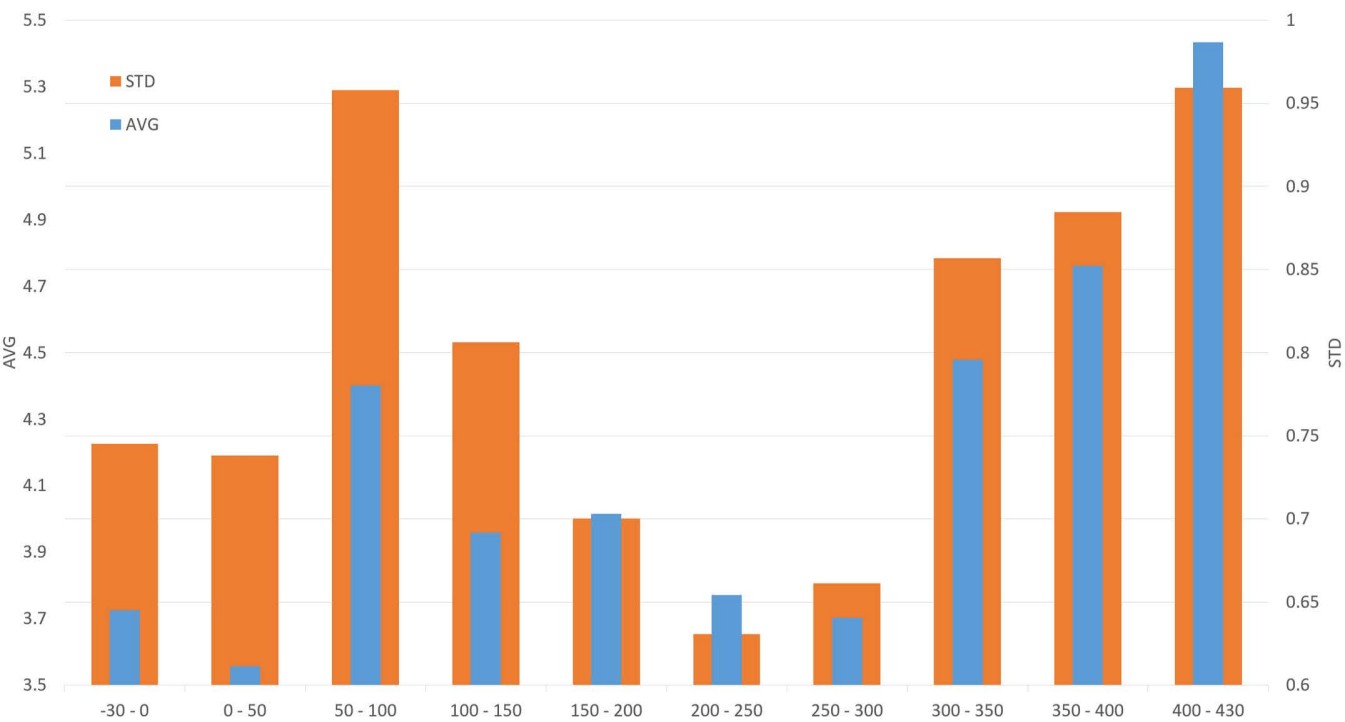

**Fig 4. Mean values (AVG; blue) and standard deviations (STD; orange) in reconstructed *Production regions* gathered in residential area locations.**

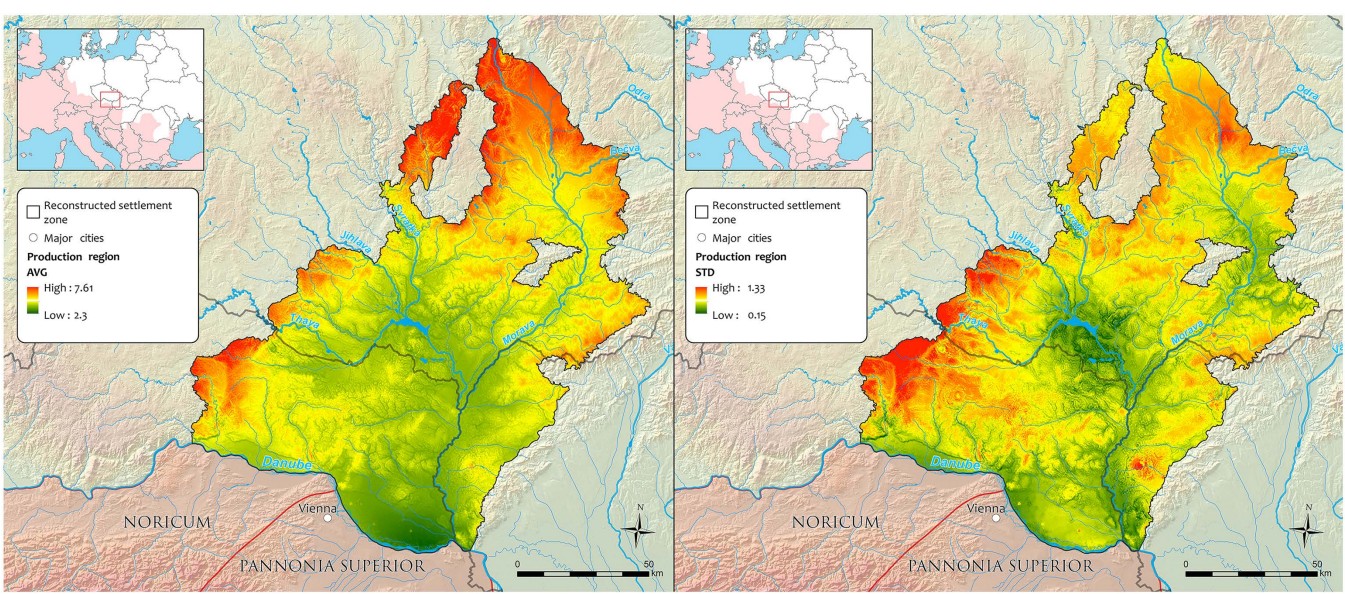

**Fig 5. Spatial distribution of the mean values (AVG) and standard deviations (STD) in reconstructed *Production regions*.**

**Fig 6. Spatial patterns of changing *Production region* expressed as differences in mean values (A) and their standard deviations (B) over the agroclimatic domain of the 'Marcomannic' settlement zone during the Roman Period.**

in both cases foreshadowed upcoming periods of decrease in quantity of archaeological data. Nevertheless, the established relationships must be understood in the context of other complex underlying processes and phenomena originating in economic, social, local and external political conditions (see Discussion). The extent, magnitude, and proportionality of their effects are reflected through the percentage of observed changes in climatic conditions and Germanic settlement structures (Table 1). Their direct association is primarily limited by the temporal resolution of archaeological data and the reflection of broader trends, leaving room for 'non-climatic' explanations of the recorded changes in Germanic residential areas.

## Discussion

The ascertained incidences in the Germanic residential area data and agroclimatic conditions need to be put into context and interpreted together with historical events and other relevant phenomena to gain a complex understanding and draw an explanation of the recorded structural changes in Germanic settlement context, particularly during the *key time blocks*. Naturally, the evidenced transition (*Foundations* and *Abandonments*) events result from complex development patterns [10] of other driving factors, such as political, social or economic decision-making (Fig 7).

The beginning of the study period is characterised by limited archaeological sources from the late La Téne Period (chronological stages D2, i.e., BCE 50–0 CE) and comparably scarce evidence for the earliest Germanic presence (stage A) in the region, originating mainly from the southern parts of the study region [10,78]. Presently, the contextual archaeological evidence of settlement activities from the stage A during the first time block (BCE 30–0), such as those available from Bohemia [79,80], are missing [78]. The archaeological data and demographic reconstructions indicate marginal Germanic activities during the earliest decades of the turn of Eras, followed by a rapid increase through the 1st century CE, reaching its peak in the first half of the 2nd century CE (Fig 2). During the second half of the 1st century CE, a short-term but significant deterioration of the agroclimatic conditions started from the late 70s, which could be potentially associated with a volcanic signal in the GISP-2 ice-cores for the year 77/78 CE (127.37 ppb) [40]. A decade later, one of the few documented periods of extensive warfare took place in the Middle Danube (the 'Suebian-Sarmatian Wars' in years 89–92 CE under Domitian and in 95 under Nerva) [81]. The conflicts known from the historical sources and so-far lacking any corroborations in the archaeological record, were attributed to the Germanic failure to provide military assistance in the Roman campaigns against Dacians. Furthermore, there is presently no convincing evidence of the direct Roman military presence on the Germanic territories of the Middle Danube [82,83], which could be attributed to the documented *Abandonments* of residential areas [10,32]. Therefore, an association with the deterioration of agroclimatic conditions could be circumstantial. Relatively scarce indications of the abnormal climatic conditions survived in available narratives for the 1st century CE [40]. Nevertheless, despite the accelerating increase of the population size through the first two centuries, the respective time block 50–100 CE exhibited only a limited amount of *Foundations* (Fig 2), which could be potentially associated with the adverse agroclimatic conditions documented during this period (Fig 4). Nevertheless, the circumstantial effect of the Roman-Germanic warfare during the time block 50–100 CE to the demographic context of the study region cannot be ruled out.

For the following 2nd century CE, the archaeological proxies indicate the highest settlement densities and the climaxing Germanic population size (Figs 2 and 6B) with its peak during the first half. The data reveal two major development shifts – the highest rate of *Foundations* in the first half and the highest *Abandonment* rates in the second half of the century. According to narrative sources, the first half of the 2nd century CE generally experienced 'calm' geopolitical conditions with predominantly peaceful political interactions [39,84] and a high level of the Roman production influx [10,31,32]. However, an isolated remark of the 'Suebian wars' during the reign of Antoninus Pius between 136 and 138 CE survived but with no further clarification or substantiation in the historical and archaeological record [85]. This period coincides with a short-term deterioration in agroclimatic conditions in documented residential areas (Fig 7A), but their causality cannot be further corroborated. Nevertheless, the exceptional reconstructed increase in the settlement *Foundations* during the time block 100–150 CE coincided significantly with the improvement of agroclimatic conditions (AVG, 90%), as well as their stability (STD, 67%) (Fig 8). Correspondingly, during this period, the Roman military presence also reached its highest numbers [85,86] to provide sufficient deterrence effect and protection.

After temporal overlap of the study period of approximately one and half century with the generally favourable conditions (Fig 3) referred to as the *Roman Climate Optimum*, presumably starting already in the 3rd century BC [1,6,40,54,87–89], a series of issues unfolded that would manifest predominantly in long-term perspective during the following 3rd century CE. The concept itself has been argued through a long-term debate regarding its general applicability and far-reaching implications to the development patterns of the Roman and other environments [66,90]. Nevertheless, the agroclimatic

**Fig 7. Environmental drivers and historical events from the Roman Period: A)** Annual AVG and STD values of the *Production regions* recorded in the residential areas (regular curves) smoothed by a 10-year filter (bold curves). **B)** The simulation of the Germanic residential areas with the probabilistic development represented through the sum of active residential areas in all simulation runs (grey) and an average number of simulated *foundation* (green) and *abandonment* (red). **C)** Reconstructed volcanic activity from Greenland ice cores (GISP-2; [55]), with events depositing > 45 ppb of sulphate highlighted in black. **D)** Selected historical events reported in the narrative sources for the Middle Danube region.

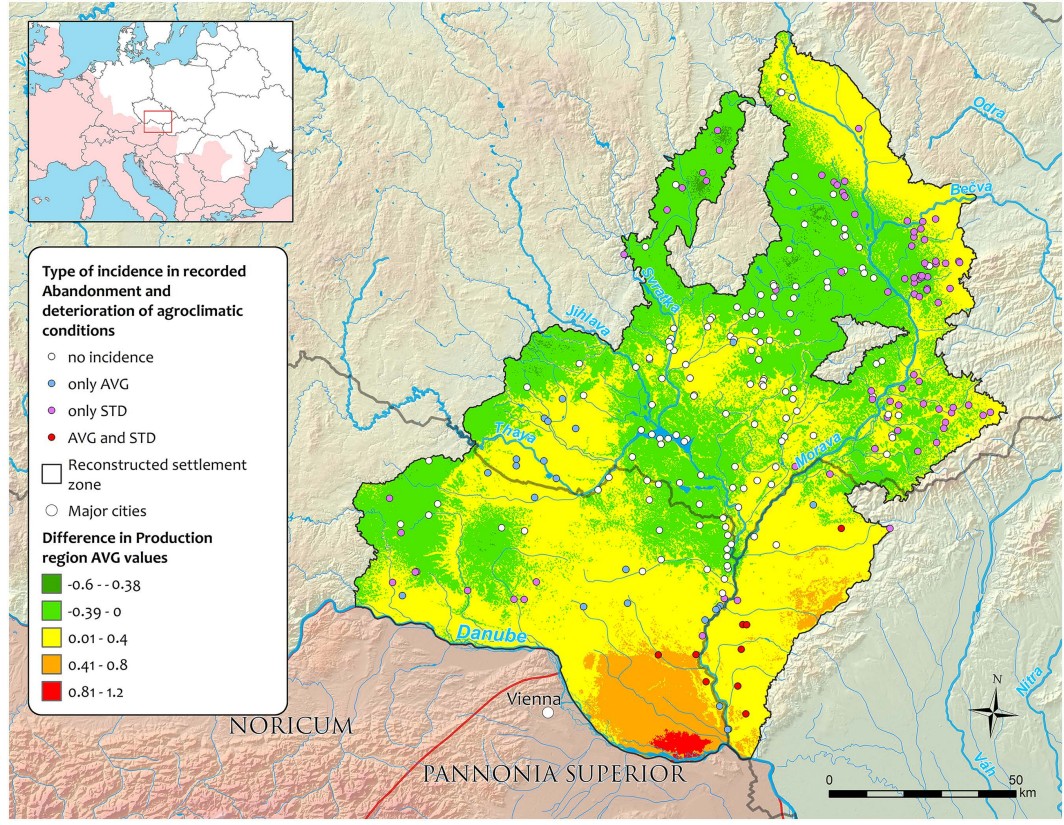

**Fig 8. The spatial patterns of production region differences between 100-150 and 150-200 CE (colour map from green to red) shown together with coinciding abandonments of the residential areas and deterioration of the agroclimatic conditions (dots differentiated by the recorded type of incidence).**

reconstructions for the study region suggest such effect foremost for the 2nd century AD. As was argued, there are the limits of the general applicability of the 'optimum' concept and interregional variability has to be considered [91,92]. Also in this study, the reconstructed agroclimatic development patterns shows a short-term deteriorations during the second half of the 1st century AD (Fig 7A).

Notably, the second half of the 2nd century CE witnessed one of the most significant Roman-barbarian conflicts in the study region and beyond – the Marcomannic Wars [8,93], which took place between 166 and 180 CE. This conflict was extensively documented in narrative sources (e.g., *SHA*, Cassius Dio, Herodianos; [42–44]), epigraphy (e.g., the Column of Marcus Aurelius and numerous inscriptions), and archaeological sources [8,25,27]. The series of barbarian incursions eventually reached northern Italy, and only with considerable effort by the Roman administration, the control of the Middle Danube province has been restored in 172 CE. Subsequently, a series of extensive military campaigns were conducted to pacify and conquer the Germanic populations in the Middle Danube territories. Among the key drivers of this turbulent epoch, the climate in the northern Germanic territories [94] is thought to have played a prominent role. Moreover, changes in the North Atlantic Oscillation (NAO) [95] may have triggered a series of migration processes.

However, the studies of archaeological and historical sources considers various intrinsic Germanic processes [8,93] (increase in population size and complexity of social structure or crisis of the subsidiary flow within the Roman-Germanic political relations (e.g., [96]), resulting in pressures in demographic conditions, redistribution system and increased mobility. The phenomena is corroborated by narrative sources, where the Germanic leaders asked to relocation of

their populations to the provincial ground [44]. The demography proxy reconstruction from archaeological data on the Germanic settlement structure would corroborate this (Figs 2 and 7B). The initial barbarian intrusions could have been also encouraged by reassignment of the Roman military units along the Danube limes for the large-scale military campaign against Parthians in 161–166 under the command of Lucius Verus. The returning Roman army has served as an effective vector of disease spread, launching the series of epidemic outbreaks throughout the Roman empire, known as Antonine plague [97].

The Greenland ice cores indicated volcanic activities in 153, 162, and 182 CE [55] with potentially exacerbating climatic instability. The physician Galenos mentioned the exceptionally severe winter of 168/169 CE in the case of recorded high plague-induced mortalities in Aquileia [98]. Along with and extensive description of the symptoms, provided by Galenos [99], such assumption also corroborate the identification of the causing pathogen of the Antonine plague with the smallpox virus, which susceptibility is positively correlated with temperature and humidity [50,97,100]. Potentially lower winter temperatures were suggested by the surviving narratives describing the Roman-Sarmatian battle on the frozen Danube during the winter of 173/174 CE [42]. It is noteworthy that the period, and particularly some of the stages of the conflict, have experienced unprecedented precipitation extremes, including four of the 20 wettest summers (171, 183, 194, and 199 CE) and even the wettest 4-year (182–186 CE) and 5-year (182–187 CE) periods during the past 2000 years [29]. Such weather extremes could have also propagated into the myth of 'miraculous rain', which role in essence might not be as much of a particular event, but rather a general contemporaneous conditions characterized by significantly increased precipitations [101].

Despite the evidence of hydroclimatic anomalies, the time block of agroclimatic aggregations did not show any marked deterioration and is represented by favourable conditions. Therefore, the high rate of settlement structure decline cannot be explained by agroclimatic changes alone, as suggested in only a limited residential area incidence of 13% in AVG and 25% in STD (Fig 8). The high number of recorded *Abandonments* could have resulted from the turbulent events connected with the Roman military presence, either through direct impacts (e.g., razing and pillaging) or indirect effects (e.g., flight prior to the Roman army advance and retreat strategy). There also have been documented changes in other segments of archaeological data suggesting a non-trivial impact of the warfare period on the Germanic societies. The derived temporal probability distributions of well-represented contexts (pit houses) find categories (e.g., brooches, metal vessels) or materials (precious metals, copper alloys, iron) show a drop between the time blocks 150–200 and 200–250 AD [10,32]. The spatial alignment of the residential areas with evidenced abandonment could provide a hint, for which most of the agroclimatically unrelated occurrences were distributed in the central low-lying parts of the region, and more significant coincidences could be seen only in the lower reaches of the Morava River and in the more peripheral areas to the east (Fig 8). It is undoubtedly challenging to distinguish and quantify the featuring drivers of the resulting *Abandonments*, but the quality of agroclimatic conditions probably played only a marginal role in this *key time block*. Another potential cause of depopulation could be a large-scale epidemic – the Antonine plague (probably smallpox [100]), which significantly impacted the Roman empire [102] and potentially influenced the Germanic territories in the Middle Danube region due to the Roman military presence [8,50].

As suggested form results of epidemiological modelling [100], providing the causing pathogen was a *variola* virus, the epidemic impact within the Roman empire would be differentiated and constrained primarily to the urban milieu as well as the regions with generally high population densities (i.e., Mediterranean coastlines, Egypt, central Italy, Achaia, or Anatolia; e.g., [65,68,71]). However, the predominantly rural regions with densities generally complying with the Iron Age societies (e.g., Danube provinces, *Gallia*; [10,22,37,103]) presumably experienced only limited impact [50,100]. Therefore, the largely less densely distributed Germanic populations of the study region with an absence of agglomerations with higher population densities [10,22], are potentially less susceptible to propagation of an epidemic outbreak [100]. Nevertheless, the presence of the Roman armies directly on the Germanic territories could have provided opportunities for disease

transition into the local populations and potential propagation of disease on the local scale [50]. However, the direct genomic confirmation of the causing pathogen is still missing for the context of the Roman Empire, as well as outside of its borders [97,99,104].

Nevertheless, the span of the Marcomannic wars of roughly 14 years (with varying intensity and geographic focus [45]) represents 28% of the respective time block 150–200 CE duration and the periods before and after the conflict could have experienced the processes which contributed to the peak of documented *Abandonments*. The non-military related processes before the Marcomannic wars, connected with the presumed immigration into the study region [8,93] or foreseeable implications to the post-war development of the Germanic settlement zone and reshaping of the power balance in Germanic milieu in general [105,106].The following time block, 200–250 CE, exhibited the most favourable agroclimatic conditions during the entire period analysed (Fig 3). These 50 years were also underlined by the second highest rate in *Foundation*, which is in firm association with the incidence of improvement of the agroclimatic conditions in individual residential areas (85% in AVG and 91% in STD). Nevertheless, the observed tendency could be also ascribed to the further 'recovery' processes of the Germanic populations during the aftermath of the Marcomannic wars [10]. Historical evidence on weather extremes remains sparse until its last decade (241–250 CE), when both the Danube and the Rhine froze regularly during the winter seasons, allowing wagons to travel on the ice [43]. A series of Roman-Germanic conflicts was also recorded between the years 233 and 236 CE during the reign of emperors Alexander Severus and Maximinus Thrax [43]. The later even realised an extensive military operation penetrating deep into the Germanic territories (Harzhorn in central Germany, Lower Saxony), propagating in archaeological record as one of two attested battlefields of the known Roman-Germanic military conflicts [107].

While it might be tempting to associate these events with the Taupo eruption (232±5 CE), one of the most significant eruptions in the past two millennia [108,109], its impact on the Northern Hemisphere seems to be limited based on the ice-core records [55] and the Middle Danube agroclimatic reconstructions (cp. Figs 7A and 6C). According to the simulation-based development of settlement structure, minor population recovery could also be anticipated after depopulation during the previous time block. Such development is often sought in influx of the Germanic populations from the Elbian cultural environment further to the west [110], observed in archaeological record foremost through the newly established extensive Late Roman Period funerary areas, such as Kostelec na Hané [111], or continuation and expansion of already existing ones [10,112,113].

During the second half of the 3rd century CE, several more significant volcanic eruptions were recorded in GIPS-2 ice cores for the years 265, 268, 283, and 286 CE (Fig 7C), with the first two reaching high amounts of sulphate (over 80 ppb). Other proxies (e.g., Greenland ice cores, Alpine tree-ring data) also suggest increased instability and deterioration of climatic conditions during the time block 250–300 CE [40]. This period witnessed a series of Germanic raids to the Middle Danube provinces in the years 258, 270–274, 282–285, and 293–299 CE [39]. Multiple factors likely contributed to these intrusions, including the limited Roman military capabilities along the Limes, constrained through the adverse effects of the '3rd-century crisis' [114,115], such as the multiple Alamannic invasions to the region of *Agri Decumates* [24] and others [114]. The processes within the Germanic milieu and potential increase of social complexity [84], increasing the pressure on the redistribution system, could also contributed to the frequencies of documented violent Roman-Germanic confrontations. Emergence of a specific funerary layer of the topmost strata of the Germanic society (e.g., Gommern, Leuna, Hassleben, Krakovany-Stráže [21,116–118]). The series of changes, apparent in this environment between the Early and the Later Roma Period (e.g., structure and scale of the funerary areas and appearance of the funerary record) certainly suggest substantial changes in social structure [10, 21, 24]. However, the marked deterioration in both quality and stability of agroclimatic conditions (AVG and STD) could have weakened the subsistence basis, adding further impetus to the barbarian raiding activities. After this period, the hydroclimatic conditions for agricultural production are about to deteriorate in a long-term perspective, foreshadowing significant climate changes and weather manifestations during the Late Antiquity in general [40,89].

However, a significant shift in the palaeoclimate record of the Late Roman Period was evident since the beginning of the 4th century CE, as the generally highly favourable agroclimatic conditions were followed by their significant deteriorations (cp. Figs 4 and 6) in quality (AVG) and stability (STD). The respective time blocks 300–350, 350–400, and 400–430 CE exhibited exceptionally high rates of recorded *Abandonments* concurrent with deteriorations in agroclimatic conditions (Table 1). An increased rate of the related narratives underlined this development trends in climatic conditions [40]. In 304 CE, a large-scale but unsuccessful invasion of the Germanic coalition across the frozen Rhine was recorded [119], which coincides with the GISP-2 record (sulphate concentration of 51.7 ppb) in 304/305 CE (Fig 7C). Isolated weather extremes were also recorded for unspecified regions of the provinces of Germania in 357 CE when early snow covered land shortly after the September equinox [120], and this year also accounted for significant barbarian intrusions into the Roman territories, including the Middle Danube region. Hostilities also broke out between 374 and 375 CE during the reign of Valentinianus I. but the conflicts are mentioned foremost as a reaction of Quadii to strengthening of the military installations along the Middle Danube and presumably also building activities on the Germanic territories [9,83]. In 368 CE, an exceptional flood of the Danube was mentioned, causing difficulties for the emperor Valens' military operations across the river [120]. This year is part of an observed more than a decade-long trend (364–374 CE) of deterioration in agroclimatic conditions, were the mean *Production region* AVG exceeds the baseline value (4.01) by 1.23, with the highest difference of 1.69 in the following year of 369 CE. Comparable short-term deteriorations are evident in 385–389 CE and 407–417 CE. Nevertheless, the Middle Danube Germanic population during these terminal development stages exhibits through the archaeological record of foremost funerary areas apparent the trend of reduction of complexity in funerary record (grave goods in burials) and 'impoverishment' [10,105,110]. It could reflect, among others, the changes in economic (the loss of connection to strategic commodity flow channels, such as Amber route) and socio-political (the shift in weight points or archaeologically documented manifestation to the neighbouring regions of Bohemia and Slovakia; [117]) alignment [10,39].

According to narrative sources, significant parts of the Middle Danube Germanic populations (Quadi amongst others are named in particular) left the area to the west in 406 CE, where another chapter of development unfolded under the so-called 'Suebian kingdom' during the Late Antiquity [121]. The archaeological record also showed a marked decrease in qualitative and quantitative aspects of the material culture and the gradual disappearance of the Germanic populations in the Middle Danube region [39,105,110].

## Conclusions

The relationship between human societies and climate encompasses various positive and negative drivers and responses. Decoding their relevance and magnitude of impact is a complex process. The testing of the anthropo-climatic responses of the Germanic populations in the Middle Danube region during the Roman Period was allowed due to the spatiotemporal intersection of two unique datasets – palaeoclimatic and archaeological, complemented by relatively ample documentary records. Despite biases in archaeological data and uncertainties in climatic reconstructions, the settlement structure closely aligns with agroclimatic conditions, demonstrating the strong influence of environmental changes on subsistence strategies. The agroclimatic context of the Germanic societies was characterised by variable conditions, ranging from generally wet and warm patterns (e.g., the first half of the 3rd century CE) to the cold and dry (e.g., the second half of the 1st century CE or the first half of the 4th century CE). Largely favourable and relatively stable conditions persisted for most of the study period until the end of the 3rd century CE when considerable long-term deterioration of agroclimatic conditions began.

For most of the *key time blocks*, reflecting more significant changes in settlement structure, increases in recorded *Foundations* were associated with favourable agroclimatic conditions and increases in *Abandonments* with their deterioration. Taking together the incidences of both types of anthropogenic responses and both AVG and STD (i.e., four dichotomy variables), only 4% of residential areas were entirely inert to the agroclimatic changes. Conversely, 18% of them exhibited a positive coincidence in both types of transitions and both respective changes in agroclimatic patterns (AVG and STD).

Such proportions substantiate the generally high dependence of the Germanic settlement structure towards environmental changes. Nevertheless, some time blocks showed low rates of settlement response to climatic variability, indicative of the susceptibility margins. Our results also suggest a higher resilience of the Germanic subsistence basis to the change of the actual quality of agroclimatic conditions (AVG, 80%) rather than its variability (STD, 84%), but differing during individual time blocks. The significant deviation from ascertained relationships shows the development during the second half of the 2nd century CE when a high number of *Abandonments* not related to climate development could have been, following archaeological theoretical models, attributed to the atrocities of the Marcomannic Wars in the Middle Danube region.

The attested incidence rates and resulting quantification of presumable climate-related changes in the study context suggest the high dependencies of these two factors. Nevertheless, these quantifications are derived from the temporal resolution through the least common denominator principle based on archaeological data. Therefore, aggregating the yearly-based agroclimatic reconstructions to the 50-year time block framework provides a relatively coarse generalization. It naturally poses a significant obstacle to differentiating the cause and consequence, which distance in time is below the available temporal resolution. Simultaneously, the climatic conditions and their implications for the primary subsistence base of the past societies represented one of many drivers of change in the complex interplay of acting factors and causal connections. As the analysis results highlighted, the agroclimatic conditions could be identified as a non-negligible forcing factor in the development of the Germanic societies of the 'Marcomannic' settlement zone. However, multiple other aspects of external (e.g., geopolitical development, Roman military activities) or internal (e.g., changes in societal complexity, availability of raw materials) nature have played a key role in shaping the resulting archaeological record with its specific biases and limits.

## Acknowledgments

The authors thanks for the comments and suggestions by academic editors and the anonymous reviewers.

## Author contributions

**Conceptualization:** Marek Vlach, Balázs Komoróczy, Max Carl Arne Torbenson, Jan Esper, Rudolf Brázdil, Ulf Büntgen, Daniela Semerádová, Otmar Urban, Jan Balek, Tomáš Kolář, Michal Rybníček, Miroslav Trnka.

**Data curation:** Marek Vlach, Daniela Semerádová, Miroslav Trnka.

**Formal analysis:** Marek Vlach, Max Carl Arne Torbenson.

**Investigation:** Marek Vlach, Balázs Komoróczy, Max Carl Arne Torbenson, Miroslav Trnka.

**Methodology:** Marek Vlach, Max Carl Arne Torbenson, Jan Esper, Rudolf Brázdil, Ulf Büntgen, Daniela Semerádová, Otmar Urban, Jan Balek, Tomáš Kolář, Michal Rybníček, Miroslav Trnka.

**Validation:** Marek Vlach, Balázs Komoróczy, Miroslav Trnka.

**Visualization:** Marek Vlach.

**Writing – original draft:** Marek Vlach.

**Writing – review & editing:** Marek Vlach, Balázs Komoróczy, Max Carl Arne Torbenson, Jan Esper, Rudolf Brázdil, Ulf Büntgen, Daniela Semerádová, Otmar Urban, Jan Balek, Tomáš Kolář, Michal Rybníček, Miroslav Trnka.

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
