## [Decision Letter · Decision Letter 0]

PONE-D-24-48398Climate variability and Germanic settlement dynamics in the Middle Danube region during the Roman Period (1st–4th Century CE)PLOS ONE

Dear Dr. Vlach,

Thank you for submitting your manuscript to PLOS ONE. After careful consideration, we feel that it has merit but does not fully meet PLOS ONE’s publication criteria as it currently stands. Therefore, we invite you to submit a revised version of the manuscript that addresses the points raised during the review process.

Your manuscript has now been seen by two referees (reviewer 1 is a co-author and was accidentally invited), whose comments are appended below. You will see from these comments that while the referees find your work of potential interest, they have raised substantial concerns that must be addressed. In light of these comments, we cannot accept the manuscript for publication, but would be interested in considering a revised version that addresses these serious concerns.

We hope you will find the referees' comments useful as you decide how to proceed. Should presentation of further data and analysis allow you to address these criticisms, we would be happy to look at a substantially revised manuscript. However, please bear in mind that we will be reluctant to approach the referees again in the absence of major revisions.

We look forward to receiving your revised manuscript.

Kind regards,

Peter F. Biehl, PhD

Academic Editor

PLOS ONE

Journal Requirements: When submitting your revision, we need you to address these additional requirements. 1. Please ensure that your manuscript meets PLOS ONE's style requirements, including those for file naming. The PLOS ONE style templates can be found at https://journals.plos.org/plosone/s/file?id=wjVg/PLOSOne_formatting_sample_main_body.pdf and https://journals.plos.org/plosone/s/file?id=ba62/PLOSOne_formatting_sample_title_authors_affiliations.pdf 2. When completing the data availability statement of the submission form, you indicated that you will make your data available on acceptance. We strongly recommend all authors decide on a data sharing plan before acceptance, as the process can be lengthy and hold up publication timelines. Please note that, though access restrictions are acceptable now, your entire data will need to be made freely accessible if your manuscript is accepted for publication. This policy applies to all data except where public deposition would breach compliance with the protocol approved by your research ethics board. If you are unable to adhere to our open data policy, please kindly revise your statement to explain your reasoning and we will seek the editor's input on an exemption. Please be assured that, once you have provided your new statement, the assessment of your exemption will not hold up the peer review process. 3. Please include your tables as part of your main manuscript and remove the individual files. Please note that supplementary tables (should remain/ be uploaded) as separate ""supporting information"" files 4. Please include captions for your Supporting Information files at the end of your manuscript, and update any in-text citations to match accordingly. Please see our Supporting Information guidelines for more information: http://journals.plos.org/plosone/s/supporting-information. 

**Additional Editor Comments:**

Your manuscript has now been seen by two referees (reviewer 1 is a co-author and was accidentally invited), whose comments are appended below. You will see from these comments that while the referees find your work of potential interest, they have raised substantial concerns that must be addressed. In light of these comments, we cannot accept the manuscript for publication, but would be interested in considering a revised version that addresses these serious concerns.

We hope you will find the referees' comments useful as you decide how to proceed. Should presentation of further data and analysis allow you to address these criticisms, we would be happy to look at a substantially revised manuscript. However, please bear in mind that we will be reluctant to approach the referees again in the absence of major revisions.

Reviewers' comments:

Reviewer's Responses to Questions

**Comments to the Author**

1. Is the manuscript technically sound, and do the data support the conclusions?

Reviewer #1: Yes

Reviewer #2: No

Reviewer #3: Yes

2. Has the statistical analysis been performed appropriately and rigorously? 

Reviewer #1: Yes

Reviewer #2: I Don't Know

Reviewer #3: Yes

3. Have the authors made all data underlying the findings in their manuscript fully available?

Reviewer #1: Yes

Reviewer #2: Yes

Reviewer #3: Yes

4. Is the manuscript presented in an intelligible fashion and written in standard English?

Reviewer #1: Yes

Reviewer #2: Yes

Reviewer #3: Yes

5. Review Comments to the Author

Reviewer #1: This interdisciplinary study is highly stimulating and most suitable for publication in PLOS1! The background and motivation are clearly outlined in the "Introduction", the data used and methods applied are described carefully, and the results are critically discussed. The study has the potential provoke further investigations at the climate-human interface, and encourage scholars from the natural and social sciences and the humanities to join forces. Although I would have written parts of the manuscript slightly differently, I recommend acceptance as it is.

Reviewer #2: The study under review entitled “Climate variability and Germanic settlement dynamics in the Middle Danube region during the Roman Period (1st–4th Century CE)” explores the relationship between climate and human societies by examining the Germanic populations in the Middle Danube region during the Roman period. While the authors utilize an interdisciplinary approach combining palaeoclimatic, archaeological, and documentary datasets, the work suffers from significant methodological and interpretive shortcomings. Below, I outline the major issues with the study.

The integration of palaeoclimatic and archaeological data is commendable, offering a valuable attempt at interdisciplinary analysis. The spatiotemporal overlap of datasets allows for a nuanced exploration of the relationship between environmental and societal factors.

However, the authors take for granted that climate significantly influenced settlement structures without critically examining the validity of this hypothesis. The study lacks an open-ended inquiry and instead seeks to quantify an assumed climate-driven relationship.

The authors narrowly attribute settlement changes to climatic variability while ignoring alternative explanations. This one-dimensional approach undermines the study’s credibility, as non-climatic factors such as warfare, economic crises, and demographic pressures likely played substantial roles. Moreover, the study conflates correlation with causation. For instance, the claim that 77% of settlements were “active” during favorable agroclimatic conditions assumes a direct causal link, ignoring other potential drivers of settlement activity.

The historical and geopolitical background of the Marcomanni in the second century CE is insufficiently addressed. Key aspects such as the societal, economic, and demographic dynamics of the Marcomannic settlement zone are mentioned but not explored in detail. The study fails to consider the impact of critical events like the Antonine Plague, shifting trade patterns, and economic disruptions in the Roman Empire, all of which likely influenced settlement dynamics.

The study uncritically adopts the concept of the “Roman Climate Optimum” and its dating from Kyle Harper’s The Fate of Rome (2017). This work, while popular, has been heavily criticized for oversimplifying complex historical and scientific issues. The term "Late Roman Period" is left undefined, leaving readers uncertain about its chronological framework.

Moreover, the study fails to situate its findings within the broader Mediterranean context. The observed settlement patterns—growth during the 1st and early 2nd centuries CE, decline in the late 2nd and early 3rd centuries, and stabilization thereafter—are not unique to the Germanic settlement zone but reflect broader trends across the Roman world. A more thorough historical contextualization could have provided critical insights into shared patterns of societal response to changing conditions.

The study’s exclusive focus on climatic drivers, lack of historical contextualization, and methodological flaws render its conclusions unconvincing. While the interdisciplinary framework holds promise, the analysis is undermined by a narrow, uncritical approach and insufficient consideration of alternative explanations. For these reasons, I recommend rejection of this study in its current form.

Reviewer #3: Very interesting paper - well conceived and executed, was fun to read. Some things to note for improvement:

1. Some English language/grammar corrections needed in the abstract, the rest is fine.

2. With regard to the archaeological data that has been synthesized (pg 5), it should be mentioned briefly by which methods the original data were obtained (e.g., intensive survey vs extensive survey, research excavation vs rescue excavation, etc.) as this can have a major effect the detection of sites and their distribution and therefore the reliability of the data.

3. Although biases and uncertainties in the archaeological and climatological data are noted, there is no acknowledgement of the possible biases in the historical documentary sources. This is worth mentioning in the text.

6. PLOS authors have the option to publish the peer review history of their article (what does this mean? ). If published, this will include your full peer review and any attached files.

**Do you want your identity to be public for this peer review?** For information about this choice, including consent withdrawal, please see our Privacy Policy .

Reviewer #1: No

Reviewer #2: No

Reviewer #3: No

---

## [Author Response · Author response to Decision Letter 1]

10 Jan 2025

Reviewer 1

This interdisciplinary study is highly stimulating and most suitable for publication in PLOS1! The background and motivation are clearly outlined in the "Introduction", the data used, and methods applied are described carefully, and the results are critically discussed. The study has the potential provoke further investigations at the climate-human interface, and encourage scholars from the natural and social sciences and the humanities to join forces. Although I would have written parts of the manuscript slightly differently, I recommend acceptance as it is.

Response: Reviewer 1 provides a positive reflection, but as a member of the author team of the paper, there is a conflict of interest, and no further reactions will be made in this regard.

Reviewer 2

The study under review entitled “Climate variability and Germanic settlement dynamics in the Middle Danube region during the Roman Period (1st–4th Century CE)” explores the relationship between climate and human societies by examining the Germanic populations in the Middle Danube region during the Roman period. While the authors utilize an interdisciplinary approach combining palaeoclimatic, archaeological, and documentary datasets, the work suffers from significant methodological and interpretive shortcomings. Below, I outline the major issues with the study.

The integration of palaeoclimatic and archaeological data is commendable, offering a valuable attempt at interdisciplinary analysis. The spatiotemporal overlap of datasets allows for a nuanced exploration of the relationship between environmental and societal factors.

However, the authors take for granted that climate significantly influenced settlement structures without critically examining the validity of this hypothesis. The study lacks an open-ended inquiry and instead seeks to quantify an assumed climate-driven relationship.

The authors narrowly attribute settlement changes to climatic variability while ignoring alternative explanations. This one-dimensional approach undermines the study’s credibility, as non-climatic factors such as warfare, economic crises, and demographic pressures likely played substantial roles. Moreover, the study conflates correlation with causation. For instance, the claim that 77% of settlements were “active” during favourable agroclimatic conditions assumes a direct causal link, ignoring other potential drivers of settlement activity.

Response: We appreciate the reflections on the manuscript. However, we feel that up to a certain proportion, the statement by the reviewer about the uncritical acceptance and unilateral endorsement of climate-driven changes of the Germanic societies of the Middle Danube region, while ignoring the other potential factors, does not entirely reflect the reality of the manuscript. On multiple occasions (lines 45-58, 320-326, 482-484), the role and potential of multiple other drivers of changes (e.g., documented military conflicts, social or economic development changes) are considered to provide unbiased insight. Particularly, the significant changes during the 2nd half of the 2nd century CE (time block 150-200 CE) cannot be associated with the agroclimatic, and the extensive conflict of the Marcomannic wars could be large-scale decreases in settlement structure size (lines 278-281, 427-442).

The study was primarily conceived to provide a deeper insight into the human-climate responses recently made available by the intersection of the two specific datasets. As a result, quantifying the potential impact of climate change on the anthropogenic context may provide substantiation for assumptions about other drivers of change. Simultaneously, the recorded co-occurrences of the documented changes in settlement structures and in agroclimatic reconstructions outside this particular warfare period are not absolute and they reach the percentages that allow for other drivers of change and forcings (lines 449-454). Nevertheless, the clues for their identification are not always apparent or present in archaeological and historical data. The objected passage in the text regarding the 77% of settlements in an ‘active’ stance during the more favourable climate conditions is provided as a general trend which has oscillating propagation in individual time periods in consideration, and it still leaves over 20% of other causes and drivers of change. The variability in incidence of climate-driven changes in several time blocks (foremost 0-50 and 250-300 CE). However, the text has been modified to provide more multilateral reflection on the potential drivers of change, whether internal or external. However, the study's findings indicate a non-accidental reliance of the Germanic subsistence on agroclimatic conditions.

The historical and geopolitical background of the Marcomanni in the second century CE is insufficiently addressed. Key aspects such as the societal, economic, and demographic dynamics of the Marcomannic settlement zone are mentioned but not explored in detail. The study fails to consider the impact of critical events like the Antonine Plague, shifting trade patterns, and economic disruptions in the Roman Empire, all of which likely influenced settlement dynamics.

Response: As suggested by the reviewer, further information on the historical context of the key period was provided (lines 397-408). The role of the Antonine Plague as one of the drivers of the upcoming ‘3rd-century crisis’ had, according to the present state of knowledge, the most significant impact within the extent of the Roman empire with high population densities, which further influenced various segments of the Roman context (economy). However, the considerably lower population densities within the Germanic study region, with the absence of agglomerations, allow only limited impact on the demography context. The spread of contagion could have been mediated by the Roman armies operating on the Germanic territories during the offensive phases of the Marcomannic wars, which are considered effective vectors of disease spread in the Roman empire.

The study uncritically adopts the concept of the “Roman Climate Optimum” and its dating from Kyle Harper’s The Fate of Rome (2017). This work, while popular, has been heavily criticized for oversimplifying complex historical and scientific issues. The term "Late Roman Period" is left undefined, leaving readers uncertain about its chronological framework.

Response: The concept of the ‘Roman Climate Optimum’ has been reflected in the study as a global underlying phenomenon, which is generally consistent with other studies, and Harper 2017 is not an exception. Yet, further references and reflections on the phenomenon were provided in the text (lines 377-385). Simultaneously, there are no further considerations on this phenomenon in the manuscript that would require further elaboration and examination.

Conversely, there is certainly a point in the necessity of providing temporal specification to the term ‘Late Roman Period’, which is embedded foremost in the chronological frameworks of the Roman Period archaeology of Germanic parts of Europe, and this deficiency was accordingly rectified in the text (lines 85-86).

Moreover, the study fails to situate its findings within the broader Mediterranean context. The observed settlement patterns—growth during the 1st and early 2nd centuries CE, decline in the late 2nd and early 3rd centuries, and stabilization thereafter—are not unique to the Germanic settlement zone but reflect broader trends across the Roman world. A more thorough historical contextualization could have provided critical insights into shared patterns of societal response to changing conditions.

The study’s exclusive focus on climatic drivers, lack of historical contextualization, and methodological flaws render its conclusions unconvincing. While the interdisciplinary framework holds promise, the analysis is undermined by a narrow, uncritical approach and insufficient consideration of alternative explanations. For these reasons, I recommend rejection of this study in its current form.

Response: It is particularly the explicit focus on the spatial domain of the intersection of the two input datasets, on which exploration is the focal point of the study set. Therefore, the comparison with the clearly heterogeneous context of the Mediterranean is not particularly relevant for the study's objectives (lines 224-226).

Reviewer 3

Very interesting paper - well conceived and executed, was fun to read. Some things to note for improvement:

1. Some English language/grammar corrections needed in the abstract, the rest is fine.

Response: The language issues in the abstract have been corrected.

2. With regard to the archaeological data that has been synthesized (pg 5), it should be mentioned briefly by which methods the original data were obtained (e.g., intensive survey vs extensive survey, research excavation vs rescue excavation, etc.) as this can have a major effect the detection of sites and their distribution and therefore the reliability of the data.

Response: This is clearly a valid point, and the text was further elaborated in the section concerning the nature and quality of input archaeological information (lines 119-129).

3. Although biases and uncertainties in the archaeological and climatological data are noted, there is no acknowledgement of the possible biases in the historical documentary sources. This is worth mentioning in the text.

Response: As in the previous point, this is also a clearly valid suggestion, and the respective section has been supplemented (lines 137-148).

---

## [Decision Letter · Decision Letter 1]

PONE-D-24-48398R1Climate variability and Germanic settlement dynamics in the Middle Danube region during the Roman Period (1st–4th Century CE)PLOS ONE

Dear Dr. Vlach,

Thank you for submitting your manuscript to PLOS ONE. After careful consideration, we feel that it has merit but does not fully meet PLOS ONE’s publication criteria as it currently stands. Therefore, we invite you to submit a revised version of the manuscript that addresses the points raised during the review process.

We look forward to receiving your revised manuscript.

Kind regards,

Peter F. Biehl, PhD

Academic Editor

PLOS ONE

Journal Requirements:

Additional Editor Comments:

Please address comments before resubmitting.

Reviewers' comments:

Reviewer's Responses to Questions

**Comments to the Author**

1. If the authors have adequately addressed your comments raised in a previous round of review and you feel that this manuscript is now acceptable for publication, you may indicate that here to bypass the “Comments to the Author” section, enter your conflict of interest statement in the “Confidential to Editor” section, and submit your "Accept" recommendation.

Reviewer #2: (No Response)

Reviewer #3: All comments have been addressed

2. Is the manuscript technically sound, and do the data support the conclusions?

Reviewer #2: Yes

Reviewer #3: (No Response)

3. Has the statistical analysis been performed appropriately and rigorously? 

Reviewer #2: Yes

Reviewer #3: (No Response)

4. Have the authors made all data underlying the findings in their manuscript fully available?

Reviewer #2: Yes

Reviewer #3: (No Response)

5. Is the manuscript presented in an intelligible fashion and written in standard English?

Reviewer #2: Yes

Reviewer #3: (No Response)

6. Review Comments to the Author

Reviewer #2: The authors have made an effort to address several of my critiques, but not all have been fully resolved. They have worked to clarify the role of non-climatic factors in settlement changes, adding references to historical events and refining their interpretation of climate’s influence. However, their response still leans heavily on climate as a primary driver, without fully engaging in a more open-ended analysis of the broader social, economic, and geopolitical forces at play. While they acknowledge alternative explanations, these remain secondary rather than being genuinely integrated into the discussion.

Their expansion on the historical and geopolitical background of the Marcomannic period, including the Antonine Plague, is a welcome addition. However, their argument that the plague had minimal impact on Germanic populations due to lower population density seems overly simplistic. While they recognize the potential role of Roman armies in spreading disease, they stop short of fully exploring how pandemics can affect frontier societies beyond direct population loss.

Regarding the use of the "Roman Climate Optimum," adding references helps provide context, but the authors do not critically engage with the debate surrounding this concept. Simply citing additional studies does not address the concern that the framework itself is contested and may oversimplify historical complexities.

Their rejection of a broader Mediterranean comparison is also unconvincing. While I understand their regional focus, the settlement patterns they describe are not unique to the Middle Danube. Acknowledging parallels elsewhere would have strengthened their argument rather than detracted from it.

Overall, while the revisions improve the manuscript in some areas, key methodological and interpretive concerns remain. The study still leans too heavily on climate determinism, and alternative explanations, though now mentioned, are not fully explored. The changes represent progress, but they do not completely resolve the fundamental issues I raised.

Reviewer #3: (No Response)

7. PLOS authors have the option to publish the peer review history of their article (what does this mean? ). If published, this will include your full peer review and any attached files.

**Do you want your identity to be public for this peer review?** For information about this choice, including consent withdrawal, please see our Privacy Policy .

Reviewer #2: No

Reviewer #3: No

---

## [Author Response · Author response to Decision Letter 2]

23 Apr 2025

Reviewer 2

The authors have made an effort to address several of my critiques, but not all have been fully resolved. They have worked to clarify the role of non-climatic factors in settlement changes, adding references to historical events and refining their interpretation of climate’s influence. However, their response still leans heavily on climate as a primary driver, without fully engaging in a more open-ended analysis of the broader social, economic, and geopolitical forces at play. While they acknowledge alternative explanations, these remain secondary rather than being genuinely integrated into the discussion.

Response:

We agree that climate only represents one part in the complex topic that is demographic changes. We also believe that much of the debate on climatic influences on Germanic and Roman population dynamics have been much too simplified due to generalisation and a lack of local (or even) regional records. However, our comparison is cemented for relatively well-defined spatial conditions. Going back 1500-2000 years, the available climate proxy records are extremely limited – especially if limiting oneself to annually-resolved estimates. As a consequence, long tree-ring chronologies from the European Alps and ice-core records from Greenland have been assumed to directly represent conditions in regions from the Mediterranean to the British Isles. The reality on the ground is likely much more complex. However, our study only compares proxy data that have been robustly verified for the past 900+ years to archaeological information from the area from which the reconstructions stem. From multiple perspectives, we believe this to be a unique situation and comparison, which aligns with previous calls (e.g., Jongman 2019) for more detailed empirical evidence and scientific archaeology. These reflections have been also incorporated to the text (e.g., lines 319-325, 446-451, 533-536, 592-599).

Reviewer 2

Their expansion on the historical and geopolitical background of the Marcomannic period, including the Antonine Plague, is a welcome addition. However, their argument that the plague had minimal impact on Germanic populations due to lower population density seems overly simplistic. While they recognise the potential role of Roman armies in spreading disease, they stop short of fully exploring how pandemics can affect frontier societies beyond direct population loss.

Response:

We acknowledge this objection as fully justified and there has been provided further elaboration on the subject reflecting the potential of the epidemic impact to the studied context (lines 456-468).

Reviewer 2

Regarding the use of the "Roman Climate Optimum," adding references helps provide context, but the authors do not critically engage with the debate surrounding this concept. Simply citing additional studies does not address the concern that the framework itself is contested and may oversimplify historical complexities.

Response:

The revised version has attempted to present a more nuanced description of the complexities the reviewer mentioned. We do not disagree with these thoughts, and although our study focuses on the agroclimatic component of variability, we do not consider it climatic determinism. Also, the concept of the ‘optimum’ does not play any substantial interpretative role in the study as the variability is more acknowledged for the regionally resolved development patterns (lines 390-397).

Reviewer 2

Their rejection of a broader Mediterranean comparison is also unconvincing. While I understand their regional focus, the settlement patterns they describe are not unique to the Middle Danube. Acknowledging parallels elsewhere would have strengthened their argument rather than detracted from it.

Response:

At this point we do not believe that the comparison is practically meaningful – in terms of cumulative uncertainty. However, this assertion should not be seen as a dismissal of the reviewer’s comment. We fully agree that Mediterranean events affected Roman strategies regarding the frontiers, but when considering that the relationship is likely lagged in time, it easily becomes an exercise in “cherry-picking” without clear quantification. Parallels with other regions (including other limes areas) suffer from the lack of high-resolution and high-quality palaeoclimatic estimates. Because of the multiple sources of uncertainty that exists in comparisons (e.g., some of the proxy records from the Mediterranean region that have previously been used lack the temporal resolution to provide meaningful basis for caparison), we argue that our approach, while acknowledging these limitations, offers one perspective of a crucial period of European history. Of course, this is not meant to be the final word on the topic. Simultaneously, further parallels to development patterns in other parts of the Germanic and Roman worlds have been supplemented (lines 222-240).

Reviewer 2

Overall, while the revisions improve the manuscript in some areas, key methodological and interpretive concerns remain. The study still leans too heavily on climate determinism, and alternative explanations, though now mentioned, are not fully explored. The changes represent progress, but they do not completely resolve the fundamental issues I raised.

Response:

We hope that the responses here and the additional changes to the manuscript will represent sufficient reflections and a compromise acceptable to all parties.

References:

Jongman, W.M. (2019) The economic archaeology of Roman economic performance. In Verhagen, P., Joyce, J., Groenhuijzen, M.R. (Eds.) Finding the Limits of the Limes. Springer, Amsterdam.

---

## [Editor Report · Decision Letter 2]

Climate variability and Germanic settlement dynamics in the Middle Danube region during the Roman Period (1st–4th Century CE)

PONE-D-24-48398R2

Dear Dr. Vlach,

We’re pleased to inform you that your manuscript has been judged scientifically suitable for publication and will be formally accepted for publication once it meets all outstanding technical requirements.

Kind regards,

Peter F. Biehl, PhD

Academic Editor

PLOS ONE
---

## [Editor Report · Acceptance letter]

PONE-D-24-48398R2

PLOS ONE

Dear Dr. Vlach,

I'm pleased to inform you that your manuscript has been deemed suitable for publication in PLOS ONE. Congratulations! Your manuscript is now being handed over to our production team.

Kind regards,

on behalf of

Dr. Peter F. Biehl

Academic Editor

PLOS ONE